

# Does novelty influence the foraging decisions of a scavenger?

Debottam Bhattacharjee[1,2,3,*], Shubhra Sau[2,4,*], Jayjit Das[2,5] and Anindita Bhadra[2]

[1] Centre for Animal Health and Welfare, Jockey Club College of Veterinary Medicine and Life Sciences, City University of Hong Kong, Hong Kong, SAR, Hong Kong

[2] The Dog Lab, Behaviour and Ecology Lab, Department of Biological Sciences, Indian Institute of Science Education and Research Kolkata, Mohanpur, West Bengal, India

[3] Department of Infectious Diseases and Public Health, Jockey Club College of Veterinary Medicine and Life Sciences, City University of Hong Kong, Hong Kong, SAR, Hong Kong

[4] Department of Botany and Zoology, Masaryk University, Brno, Czech Republic

[5] Department of Endangered Species Management, Wildlife Institute of India, Dehradun, Uttarakhand, India

[*] These authors contributed equally to this work.

Corresponding author
Anindita Bhadra,
abhadra@iiserkol.ac.in

## ABSTRACT

Acquiring knowledge about the environment is crucial for survival. Animals, often driven by their exploratory tendencies, gather valuable information regarding food resources, shelter, mating partners, *etc.* However, neophobia, or avoiding novel environmental stimuli, can constrain their exploratory behaviour. While neophobia can reduce potential predation risks, decreased exploratory behaviour resulting from it may limit the ability to discover highly rewarding resources. Dogs (*Canis familiaris*) living in semi-urban and urban environments as free-ranging populations, although subject to various selection forces, typically have negligible predation pressure. These dogs are scavengers in human-dominated environments; thus, selection against object-neophobia can provide benefits when searching for novel food resources. Although captive pack-living dogs are known to be less neophobic than their closest living ancestors, wolves (*Canis lupus*), little is known about free-ranging dogs' behavioural responses to novel objects, particularly in foraging contexts. Using an object choice experiment, we tested 259 free-ranging dogs from two age classes, adult and juvenile, to investigate their object-neophobia in a scavenging context. We employed a between-subject study design, providing dogs with a familiar and a potentially novel object, both baited with equal, hidden food items. Adult and juvenile dogs significantly inspected the novel object first compared to the familiar one, even when the hidden food item was partially visible. To validate these findings, we compared novel objects with different strengths of olfactory cues (baited *vs.* false-baited) and found that they were inspected comparably by adults and juveniles. No significant differences were found in the latencies to inspect the objects, suggesting that free-ranging dogs may still be cautious when exploring their environments. These results indicate that free-ranging dogs, evidently from an early ontogenetic phase, do not show object-neophobia, as demonstrated by their preference for novel over familiar food sources. We conclude that little to no constraint of neophobia on exploratory behaviour in semi-urban and urban-dwelling animals can guide foraging decision-making processes, providing adaptive benefits.

## INTRODUCTION

Knowledge about the immediate environment is essential for survival. Animals navigate their surroundings and gather crucial information on food sources, mating partners, shelter, and predators (*Mettke-Hofmann, Winkler & Leisler, 2002*; *Dall et al., 2005*; *Moretti et al., 2015*; *Sarkar & Bhadra, 2022*), often driven by their exploratory behaviour. An enhanced exploratory behaviour may further lead to innovative problem-solving (*Wat, Banks & McArthur, 2020*; *Klump et al., 2022*). Thus, exploratory behaviour directly or indirectly influences the survival and reproduction of animals. However, neophobia, or avoiding objects or other environmental aspects, such as space, food, *etc.*, solely because they have never been experienced, can constrain exploratory behaviour (*Stöwe et al., 2006a*). While a suppressed tendency to explore can decrease the risk of encountering predators, it can also substantially limit opportunities to discover novel resources, like food (*Stöwe et al., 2006b*). Animals inhabiting urban and semi-urban (*i.e.*, human-dominated) environments experience a lower predation pressure than those in rural and wild habitats (*Fischer et al., 2012*; *Eötvös, Magura & Lövei, 2018*). Therefore, reduced object-neophobia in urban-dwelling animals can enhance exploratory behaviour, providing benefits (but see, *Meddock & Osborn, 1968*). A long-standing bias of scientists, especially behavioural ecologists, has been to ignore wildlife in urban environments (*Magle et al., 2012*), particularly 'subsidised' animals that 'exploit' anthropogenic food resources. Consequently, our knowledge of these animals' neophobic behaviour is obscured. In recent years, with the formalisation of urban ecology, scientific research has developed more interest in understanding the behavioural and cognitive aspects of decision-making of animals living close to humans.

Human-dominated environments impose novel challenges (*Ditchkoff, Saalfeld & Gibson, 2006*), which can be perceived differently by animals that are domesticated, non-domesticated synanthropes, and non-domesticated but coerced to live around humans (*Beckman, Richey & Rosenthal, 2022*). While the extent of damage by the Anthropocene is irrevocable (*Wagner et al., 2021*), a wide range of animals exhibit behavioural adaptations and plasticity in traits to benefit from human-induced rapid environmental changes (*Sih et al., 2010*). Reduced object-neophobia and enhanced exploratory behaviour are such behavioural adaptations (*Griffin, Netto & Peneaux, 2017*). However, it is still unresolved to what extent neophobia is repeatable (*i.e.,* a personality trait) or plastic, due to inconsistency in definition and objective assessments (*Takola et al., 2021*; *Kimball & Lattin, 2023*), but see, (*Day et al., 2003*; *Quinn et al., 2009*; *Grunst et al., 2019*). The problem increases further when neophilia, or the preference for novelty over familiarity (*Day et al., 2003*), is considered the opposite of neophobia. Although neophilia can promote exploratory behaviour (*Day et al., 2003*), object-neophobia and neophilia are thought to be shaped by different selective forces, and they do not necessarily represent two extremes of a continuum (*Greenberg & Mettke-Hofmann, 2001*). Most importantly, neophobic and neophilic responses are highly context-specific, with studies typically testing neophilia in neutral and neophobia in non-neutral, such as foraging contexts (*Takola et al., 2021*). Empirical evidence suggests that urban-living animals have reduced object-neophobia (*Tryjanowski et al., 2016*; *Greggor et al., 2016*; *Jarjour et al., 2019*; *Biondi et al., 2020*;

*Miller et al., 2022*, but see *Mazza et al., 2021*), which, with or without the presence of neophilia, can drive their enhanced exploratory behaviour (*Griffin, Netto & Peneaux, 2017*; *Thompson et al., 2018*; *Breck et al., 2019*; *Dammhahn et al., 2020*). Despite a wealth of knowledge gathered by empirical studies on these traits, our understanding of the ecologically relevant contexts under which such traits can be expressed and/or be plastic is lacking (see *Gordon, 2011*). Consequently, testing more species—that live in close proximity to humans—in ecologically relevant contexts, such as foraging, is imperative.

Dogs (*Canis familiaris*) inhabit a wide range of habitats, from living as pets in human households to roaming freely in human-dominated environments. Unlike pets, free-ranging dogs are primarily under natural and sexual selection pressures (*Range & Marshall-Pescini, 2022*); thus, acquiring information about their surroundings is key to survival. Several recent studies have provided evidence that free-ranging dogs' socio-cognitive abilities, from understanding human cues, attentional states and facial expressions to forming trust with and learning socially from humans, have made them successful in human-dominated environments (*Bhattacharjee et al., 2017b*; *Bhattacharjee et al., 2017c*; *Bhattacharjee et al., 2019*; *Bhattacharjee, Sau & Bhadra, 2018*; *Bhattacharjee, Sau & Bhadra, 2020*; *Brubaker et al., 2019*; *Bhattacharjee & Bhadra, 2020*; *Bhattacharjee & Bhadra, 2022*; *Lazzaroni et al., 2023*; *Cimarelli et al., 2023*). However, in contrast to socio-cognitive skills, little is known about individual traits, such as exploratory behaviour and their contribution to free-ranging dogs' success in human-dominated environments. These dogs spend considerable time and energy walking and foraging solitarily (*Sen Majumder, Chatterjee & Bhadra, 2014*), which could potentially include encountering unknown human artefacts, and if that explored, can be rewarding (such as packaged containers, garbage bins, *etc.*, D Bhattacharjee, A Bhadra, pers. obs., 2010–2024). Therefore, reduced object-neophobia can be beneficial for free-ranging dogs. Pet and captive pack-living dogs exhibit little to no neophobia and enhanced neophilia (*Kaulfuß & Mills, 2008*; *Moretti et al., 2015*) in choice tasks when provided with familiar and unfamiliar objects. The presence of group members in captive pack-living dogs further induces exploratory behaviour, indicating risk-sharing (*Moretti et al., 2015*). However, considering early age classes (five, six, and eight weeks of age), a dog-wolf comparative study suggests that wolves are more persistent in exploring a novel environment and novel objects when compared with dogs (*Marshall-Pescini et al., 2017a*). While various scavenging strategies of free-ranging dogs have been reported in the past, from following simple rules of thumb to extractive foraging techniques (*Mangalam & Singh, 2013*; *Bhadra et al., 2015*), how these dogs respond to novelty during foraging and whether their exploratory behaviour and foraging decisions are constrained or influenced by object-neophobia is unknown.

We conducted an ecologically relevant object-choice test with free-ranging dogs to investigate whether their foraging decisions depend on (or are constrained by) object-neophobia. Free-ranging dogs encounter plastic garbage bags and pouches and extract food leftovers from them (*Bhattacharjee et al., 2017a*). On the contrary, plastic balls are unlikely to be encountered during scavenging (D Bhattacharjee, A Bhadra, pers. obs., 2010–2024). Accordingly, we used plastic balls as novel and plastic pouches as familiar objects. A between-subject study design was employed, and dogs were randomly provided

with one of the following conditions with specific combinations of objects—an opaque ball and an opaque pouch (*test*), an opaque ball and a translucent white pouch (*test*), an opaque pouch and a translucent white pouch (*control*), and two opaque balls (*validation*). All the objects contained equal-sized food items, except for the last condition (*i.e.,* two opaque balls), in which of one the balls was false-baited. Since free-ranging dogs benefit from reduced intraspecific competition during scavenging and experience little to no predation pressure in human-dominated environments (*Sarkar, Sau & Bhadra, 2019*; *Sarkar et al., 2023*), we hypothesised that they would not exhibit object-neophobia. In particular, we expected dogs to inspect the novel objects first rather than the familiar ones in the test conditions but show comparable choices in the control and validation conditions. Since object-neophobia might have been selected against during domestication (cf. *Moretti et al., 2015*), we expected adults and juveniles to respond similarly, *i.e.,* no ontogenetic differences in their behavioural responses. Additionally, as humans exert significant anthropogenic stress on free-ranging dogs (*Bhattacharjee & Bhadra, 2020*; *Bhattacharjee & Bhadra, 2021*) and even cause mortality (*Paul et al., 2016*), we hypothesised that dogs would be cautious in their approach to the objects. Thus, we expected them to show comparable latencies to approach the familiar and novel objects first under different conditions.

## METHODS

**(i) Ethical statement:** In India, free-ranging dogs are protected by the Prevention of Cruelty to Animals Act (1960) of Parliament, which allows interactions with dogs, including feeding and petting. Our study adhered to the guidelines of the act and the ethical guidelines of animal testing of the Indian Institute of Science Education and Research Kolkata (Approval no. 1385/ac/10/CPCSEA). The subject dogs were tested in their natural habitats, and their participation in the tests was voluntary. All meat items used were fresh and fit for human consumption.

(ii) **Study area and subjects:** The study sites were the following semi-urban and urban areas of West Bengal, India - Mohanpur (semi-urban, 22.946944 and 88.534444), Kalyani (semi-urban to urban, 22.975, 88.434444) and Kolkata (urban, 22.957222, 88.610833). We covered a total sampling distance of 137 Km (Mohanpur: 18 Km, Kalyani: 27 Km, and Kolkata: 92 Km). The study was conducted from October 2016 to May 2017, between 9 AM and 6 PM. The experimenters walked on the streets to locate dogs, preferably present without group members. In case more than one dog was present, a focal subject dog was chosen haphazardly and lured out of sight of the group members. Portions of this text were previously published as part of a preprint (https://doi.org/10.1101/2023.12.12.571372).

We tested 274 free-ranging dogs from two age classes: adults ($n = 147$) and juveniles ($n = 127$). While the exact age of the dogs was unknown, morphological characteristics enabled us to define an age class precisely (*Sen Majumder et al., 2014*). The sexes of the dogs were noted by visually inspecting their genitalia. Of the 147 adult dogs, 77 were females, whereas 63 out of 127 were females in the juvenile age class. Thus, the male–female ratio of our overall sample was close to 1:1. Since tracking a large sample of dogs over the experimental period is highly challenging, we decided to use a between-subject study

design. All dogs were tested once, and to rule out any potential bias of resampling, we tested dogs from different locations and never revisited the same area for testing. Additionally, we relied on morphological characteristics, such as coat colour, ear shape, and body size of the dogs from the videos to ensure that the same individual was not re-tested.

**(iii) Experimental objects and food items:** We used three different objects in this study - a green opaque plastic ball with a diameter of 7 cm, a black opaque plastic pouch of 19 cm × 11 cm size, and an identically sized white translucent plastic pouch. As dogs have dichromatic colour vision (*Siniscalchi et al., 2017*), we chose the colour green to avoid any potential effect of chromaticity. Additionally, since the experiments were conducted outdoors (with other green environmental objects, like grass, shrubs, trees, etc. around), we expected the novel objects to blend well with the environment. Each plastic pouch had a 0.12 mm thickness. The contents placed inside the opaque objects were not visible from the outside. However, the white translucent plastic pouch provided a partial visual cue of its contents without revealing it completely. We used raw chicken pieces (∼15 g) as food items. A circumferential opening was made in the plastic ball to insert the food item. Later, the opening was loosely closed using transparent tape such that the structure of the ball remained intact. Similarly, after placing the food item inside, the opening of the plastic pouch was loosely tied with cotton threads. As the balls and pouches were not tightly closed, we expected them to provide similar olfactory cues to dogs. For each experimental trial, new and completely unused objects were used.

**(iv) Experimental conditions and procedure:** Upon locating an adult or a juvenile dog, the experimenter randomly presented them with one of the following experimental conditions with a specific combination of objects:

(a) Ball and opaque pouch: Test condition using novel and familiar objects with similar visual obscurity (*i.e.,* opacity). Both objects had food items inside, thus providing similar olfactory cues. This condition tested preference between novel and familiar objects with similar visual obscurity and olfactory cues of the food items (Video S1).

(b) Ball and translucent white pouch: Test condition using novel and familiar objects with different visual obscurity. Both objects had food items inside, thus providing similar olfactory cues. This condition tested preference between novel and familiar objects when the familiar object provided partial visual cues of the food item (Video S2).

(c) Opaque pouch and translucent white pouch: Control condition using familiar objects with different visual obscurity. Both objects had food items inside, thus providing similar olfactory cues. This condition tested whether familiar objects with similar olfactory cues were perceived comparably or discriminated against based on visual cues (Video S3).

(d) Ball and scent ball: Validation condition using novel objects with similar visual obscurity. One of the balls had a food item inside, similar to the first two conditions. In contrast, the other ball (*i.e.,* scent ball) was false-baited by gently rubbing its inside with a food item. Therefore, the novel objects varied in their strengths of olfactory cues. This condition checked if olfactory cues determined object choice over novelty (Video S4).

The two objects were placed on the ground approximately 1 m apart, regardless of conditions. We used a pseudorandomised order to place the objects (left/right) to avoid any potential effects of side bias. The objects were equidistant from a focal dog. The minimum

distance between the midpoint of objects and the focal dog was approximately 2 m. The experimenter stood 0.5 m behind the midpoint of the two objects and tried to get the attention of a focal dog by calling "aye aye" (*Bhattacharjee et al., 2017c*). After capturing the dog's attention (*i.e.,* when the focal dog's head was oriented towards experimental setup), the experimenter left the set-up and positioned himself at a minimum distance of 5 m away or hid behind a tree or car. This step ensured the participation of relatively shy individuals and eliminated any potential influence of human presence and subsequent begging-related behaviour exhibited by the dogs. A trial began immediately after capturing the focal dog's attention and lasted 60 s. If a dog was not attentive, the experimenter made another attempt after 10 s. A maximum of two such additional attempts were made before terminating a trial. The trials were video recorded by a person other than the experimenter from a minimum distance of 5 m using a handheld camera. S.S. and J.D., both males of similar height and physical build, played the roles of the experimenter. Therefore, dogs' responses were unlikely to be influenced by the two different experimenters involved in the study. An effect of experimenter was also unlikely because the dogs witnessed them only briefly.

**(iv) Data coding:** We coded two behavioural variables—object choice (or object preference) and the latency to choose the first object. S.S. coded the videos using a frame-by-frame video inspection method. Another rater coded 15% of the videos to check for reliability. We investigated the intraclass correlation coefficient (ICC) values from the intraclass correlation tests to assess reliability. Reliability was found to be excellent (first inspection: (ICC (3,k)) = 0.99, $p < 0.001$; latency - (ICC (3,k)) = 0.94, $p < 0.001$). Object choice was defined as the first physical inspection of an object by touching (with the muzzle) or licking (with the tongue). Thus, a clearly visible physical interaction between dogs and the objects was considered. We noted which object was inspected first (*i.e.,* ball or opaque pouch, ball or translucent white pouch, opaque pouch or translucent white pouch, and ball or scent ball) in all four conditions. Latency was defined as the time (in seconds) a focal dog took to inspect the first object from an initial 2 m distance. Thus, latencies were measured within conditions for the objects inspected first. Due to the potentially varying difficulty levels associated with food retrieval from the objects (including the movement of the ball), we decided not to code object-based activities after the first inspection. Although object-based activities may provide information on exploratory behaviour and foraging decision-making, object choice by first inspection in itself incorporates the primary crucial step of foraging decision-making. Of the 274 dogs, 15 (nine adults and six juveniles) did not move from their initial position. Therefore, we conducted our analyses on a revised sample of 259 free-ranging dogs.

**(v) Statistics:** All statistical analyses were conducted using R version 4.3.1. (*R Core Team, 2020*). We used binomial tests for the four experimental conditions separately to check whether dogs preferred one object over another. Generalised linear models (GLM) were used to investigate if age class (adult and juvenile) influenced the decision of object choice. Four binomial GLMs were run, each one for the different experimental conditions, where the type of object and age class were included as response and independent variables, respectively. In addition, we added the sexes of the dogs (male and female) as control

variables in those models. We used linear effect models (LM) to investigate the latency of first inspection. Latency was included as the response variable, and age class, sex (as a control variable), and the corresponding object inspected were included as fixed effects in the models. We ran four LMs for the different experimental conditions. GLM and LM were conducted using the "lme4" package (*Bates et al., 2015*). If full models had significant effects, comparisons with null models were checked using the "lmtest" package (*Hothorn & Zeileis, 2011*). Model diagnostics were checked using the "DHARMa" package (*Hartig, 2020*). If model residuals violated normality assumptions, we log-transformed the response variable and re-ran the model. The level of statistical significance ($\alpha$) was set at 0.05.

## RESULTS

### (i) First inspection (object choice)

(a) Ball and opaque pouch—62% (23 out of 37) of the adults and 69% (18 out of 26) of the juveniles inspected the ball first. Regardless of age classes, dogs first inspected the ball significantly more over the opaque pouch (Binomial test: $p = 0.02$, 95% Confidence Interval/CI [0.52–0.76], Fig. 1). However, we found no significant effect of age class (GLM: $z = 0.6$, $p = 0.54$) and sex (GLM: $z = 0.3$, $p = 0.76$) on object choice. These results suggest that both adults and juveniles comparably preferred the novel object over the familiar one.

(b) Ball and translucent white pouch—73% (25 out of 34) of the adults and 65% (17 out of 26) of the juveniles inspected the ball first. Overall, dogs first inspected the ball significantly more than the translucent white pouch (Binomial test: $p = 0.002$, 95% CI [0.57–0.81], Fig. 1). Age class (GLM: $z = -0.68$, $p = 0.49$) and sex (GLM: $z = 1.12$, $p = 0.26$) did not influence object choice. Thus, similar to the ball and opaque pouch condition, adults and juveniles first inspected the novel object more than the familiar one, even when the familiar object provided visual cues of the food item.

(c) Opaque pouch and translucent white pouch—52% (15 out of 29) of the adults and 63% (19 out of 30) of the juveniles inspected the opaque pouch first. In general, dogs did not differ in their first inspection of the familiar objects (Binomial test: $p = 0.29$, 95% CI [0.44–0.70], Fig. 1), even when they differed in visual obscurity levels. No significant effect of age class (GLM: $z = 0.78$, $p = 0.43$) and sex (GLM: $z = -0.5$, $p = 0.61$) on object choice was found. (d) Ball and scent ball –56% (22 out of 39) of the adults and 54% (21 out of 39) of the juveniles first inspected the ball. Overall, dogs did not discriminate between the novel objects during the first inspection, even when the strengths of olfactory cues differed (Binomial test: $p = 0.49$, 95% CI [0.42–0.65], Fig. 1). We did not find any significant effect of age class (GLM: $z = -0.33$, $p = 0.74$) and sex (GLM: $z = -0.5$, $p = 0.61$) on object choice.

### (ii) Latency of first inspection

(a) Ball and opaque pouch—The latencies to first inspect the ball (mean $\pm$ standard deviation: $2.90 \pm 3.57$ s) and the opaque pouch ($2.13 \pm 1.93$ s) did not differ (LM: $t = 1.27$, $p = 0.20$). We found no effect of age class (LM: $t = -1.83$, $p = 0.07$) and sex ($t = -1.28$, $p = 0.20$) on latencies.
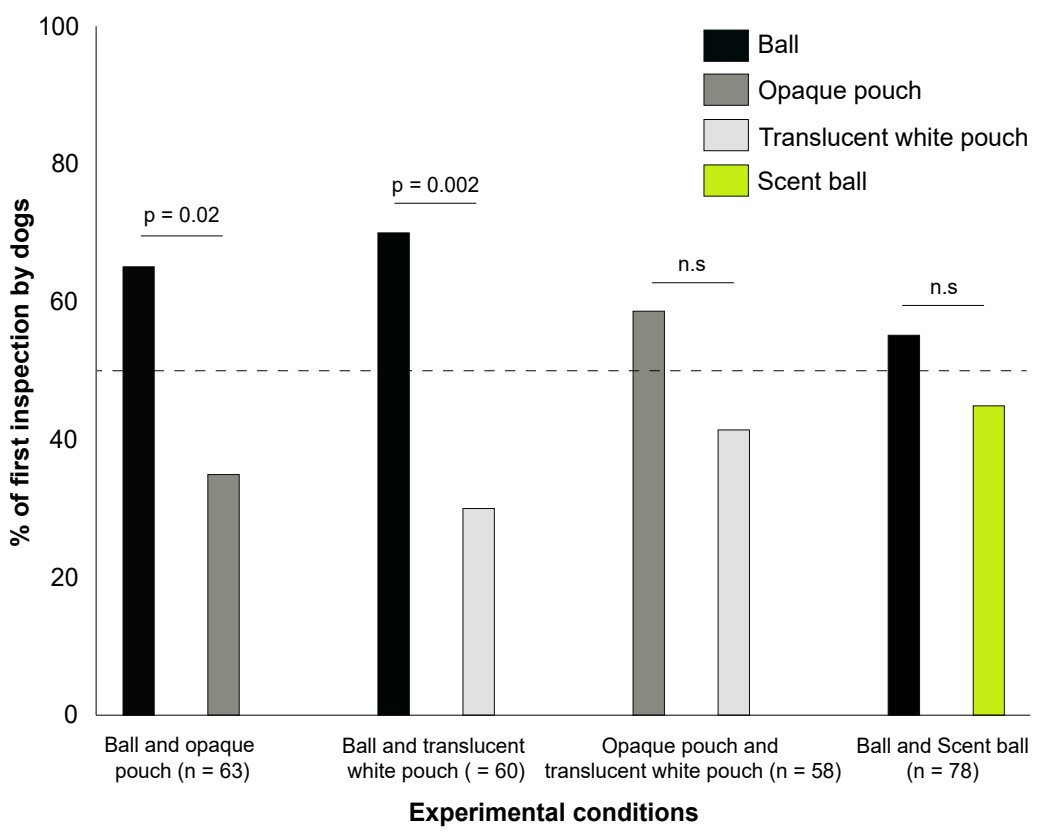

**Figure 1** **The percentage of first inspection by dogs presented with different paired alternative stimuli.** The percentage of first inspection by dogs presented with different paired alternative stimuli. *P*-values are shown for treatments that were significantly different, and non-significant differences are denoted by "n.s".

(b) Ball and translucent white pouch—Dogs first inspected the ball ($2.42 \pm 1.51$ s) comparably to the translucent white pouch ($4.05 \pm 6.43$ s) (LM: $t = 0.91$, $p = 0.37$). No effect of age class ($t = -1.13$, $p = 0.26$) and sex ($t = 0.13$, $p = 0.89$) on the latencies was found.

(c) Opaque pouch and translucent white pouch—The latencies to first inspect the opaque pouch ($2.11 \pm 1.57$ s) and the translucent white pouch ($3.16 \pm 4.32$ s) were similar (LM: $t = 1.20$, $p = 0.23$). Age class ($t = 1.48$, $p = 0.14$) and sex ($t = 1.33$, $p = 0.18$) did not impact the latencies of first inspection.

(d) Ball and scent ball—We did not find any difference in dogs' latencies to first inspect the ball ($2.67 \pm 2.17$ s) and the scent ball ($2.31 \pm 1.15$ s) (LM: $t = -1.10$, $p = 0.27$). No effect of age class ($t = 0.09$, $p = 0.93$) and sex ($t = -1.15$, $p = 0.25$) was found.

## DISCUSSION

Using an ecologically relevant experimental design, we investigated whether free-ranging dogs exhibit object-neophobia and whether their foraging decision-making is dependent on, and in particular, constrained by, neophobic behaviour. We tested dogs from two

age classes to evaluate if such behavioural responses have any ontogenetic developmental basis. As hypothesised, dogs, irrespective of age classes, first inspected the ball significantly more than the opaque pouch. Similarly, adults and juveniles inspected the ball significantly more than the translucent white pouch. We further investigated dogs with balls of varying strengths of olfactory cues and found no significant difference in their first inspection. Within the four conditions, we found no significant differences in latencies between dogs who first inspected the two objects (*i.e.,* dogs who inspected the ball and dogs who inspected the opaque pouch, dogs who inspected the ball and dogs who inspected the translucent white pouch, dogs who inspected the opaque pouch and dogs who inspected the translucent white pouch, dogs who inspected the ball and dogs who inspected the scent ball). We discuss these findings and their implications, emphasising object-neophobia, exploratory behaviour and foraging decision-making in free-ranging dogs.

Our results indicate that free-ranging dogs do not exhibit object-neophobia in a scavenging context, as demonstrated by their first inspection of the ball over familiar plastic pouches. This implies that object-neophobia has no direct constraint on their potential exploratory behaviour. While we did not investigate the exploration of objects after the first inspection, object choice in the form of first inspection highlights a crucial step of exploratory behaviour and foraging decision-making (*Mettke-Hofmann et al., 2006*; *Takola et al., 2021*). These findings align with the conclusions from previous studies, which suggest a reduced neophobia in dogs, albeit in non-foraging contexts (*Kaulfuß & Mills, 2008*; *Moretti et al., 2015*). Human-dominated environments offer food subsidies, and these predictable resources can have broad ecological and evolutionary implications for animals (*Oro et al., 2013*). A wide range of species exploit these resources (*Biswas et al., 2023*), resulting in substantial competition both within and between species. Furthermore, anthropogenic activities can alter the spatiotemporal availability of such resources and directly or indirectly determine how animals eventually utilise them (*Markus & Hall, 2004*; *Murray & St. Clair, 2017*; *Ramírez et al., 2020*; *Bhattacharjee & Bhadra, 2021*; *Egert-Berg et al., 2021*). To maximise their food intake while avoiding potential anthropogenic stressors, animals may engage in opportunistic scavenging and explore novel resources, including objects. A reduction in object-neophobia in free-ranging dogs can, therefore, be an adaptive strategy to promote exploratory behaviour, especially during foraging.

A comparable response of adults and juveniles can be attributed to selection against neophobia during domestication, where reduced object-neophobia can provide adaptive benefits (*Moretti et al., 2015*) in human-dominated environments. In free-ranging dogs, different ontogenetic phases hold varying degrees of significance. The juvenile phase of development (3–6 months) involves complete independence from the mothers and venturing into the immediate environment (*Paul et al., 2016*); developmentally, this is a crucial phase as dogs start to forage on their own and may experience novel environmental aspects, like potentially rewarding novel objects. It has been shown that juvenile free-ranging dogs are reluctant to approach and follow the communicative intents of unfamiliar humans (*Bhattacharjee et al., 2017b*). Therefore, it can be assumed that scavenging, but not begging for food from humans, is the predominant feeding strategy in juveniles. A reduced object-neophobia can thus provide considerable foraging benefits to juveniles.

Conversely, adults can rely on scavenging and begging (*Sen Majumder et al., 2014*; *Bhadra et al., 2015*; *Boitani et al., 2016*), and depending on the energy requirements and other external factors (such as human disturbance), they may flexibly switch between those strategies, where reduced object-neophobia would still be beneficial. Nonetheless, object-neophobia in dogs appears to be a trait that was selected against during domestication (but see *Marshall-Pescini et al., 2017a*). The 'social-ecology' hypothesis suggests that feeding ecology and social organisation may act together as mechanisms to drive dogs' interactions with environmental, particularly novel stimuli, including objects (*Marshall-Pescini et al., 2017b*). Free-ranging dogs' scavenging behaviour in human-dominated environments, thus, can contribute to the reduction in object-neophobia.

The first inspection of the novel object, overriding the partial visual cue of food item from the familiar object, can be attributed to free-ranging dogs' reliance on a complex multimodal sensory information system of vision and olfaction during foraging. Free-ranging dogs are primarily known to rely on olfactory cues to make their foraging decisions (*Bhadra et al., 2015*; *Banerjee & Bhadra, 2019*; *Sarkar et al., 2023*). Our results indicate that they might also use visual cues and rely more on them than olfaction when encountering potentially novel objects. In non-scavenging conditions, dogs have been shown to tolerate human artefacts (*Kaulfuß & Mills, 2008*). Thus, it is safe to assume that dogs paid particular attention to the visual appearance of the novel object. At the same time, they did not do so for the familiar objects with different visual obscurities. On the other hand, the ball and scent ball condition validates our results by demonstrating that novelty indeed played a role, and dogs did not fully rely on (the strength of) olfactory cues. These findings indicate that dogs can use their multimodal sensory information system rather flexibly, depending on the context, to guide their foraging decisions.

The appearances of novel objects, from simple to relatively 'complex' sizes and shapes, can be perceived differently by animals, eliciting varying responses (*Mettke-Hofmann et al., 2006*). In the current study, we could not measure the construct validity due to the use of only one type of novel object (*Greggor, Thornton & Clayton, 2015*; *Kimball & Lattin, 2023*), and to what extent the plastic ball was novel could not be assessed. We were cautious about calling it a truly novel object and hence considered it 'potentially novel'. It is still possible that some of the dogs might have seen plastic balls of different sizes or colours previously if they did not interact with them. Furthermore, a recent study has shown a strong preference for the colour yellow in free-ranging dogs in the context of foraging. This preference could be strong enough to override the attraction towards food rewards (*Roy et al., 2024*). Since our plastic balls were green in colour, they would have appeared a light shade of yellow to the dogs, thereby potentially adding an additional cue to the object. Although non-significant, it is worth mentioning that a substantial percentage of adults and juveniles first inspected the familiar plastic pouches. A few possibilities can explain the finding: first, these dogs did not perceive balls as novel objects (see *Mettke-Hofmann et al., 2006*); second, the state of hunger might have forced dogs to choose the plastic pouches as reliable food sources over the balls (see *Alley, 2018*); third, these individuals had shy or avoidant personalities (see *Sloan Wilson et al., 1994*). From our current study with one-off experiments, it is impossible to pinpoint which mechanism played the most significant role.

Nevertheless, we believe that it would be an oversimplification to assert that free-ranging dogs have reduced object-neophobia, and thus, we recommend careful assessments of the highlighted mechanisms to reach a firm conclusion.

Studies with novel object tests often consider investigating the time animals take to approach or explore the objects. Individuals with reduced neophobia are expected to approach the novelty quicker than their neophobic counterparts (*Takola et al., 2021*; *Kimball & Lattin, 2023*). One may argue that dogs in our study did not differ in their latencies to first inspect the novel and familiar objects. This could be explained by the study design, where we provided two-way object choice conditions instead of separately presenting novel or familiar objects. The findings are also in line with our prediction that although free-ranging dogs may show reduced object-neophobia, the negative impact of anthropogenic stressors (*Paul et al., 2016*; *Bhattacharjee & Bhadra, 2020*; *Bhattacharjee & Bhadra, 2021*) may force them to explore their environments with caution (*Greenberg, 2003*). Besides, solitary foraging is prevalent in free-ranging dogs; consequently, a lack of intraspecific competition may aid in that process. However, this does not disregard the potential influence of competition and subsequent shorter latencies to approach novel objects during group foraging events.

## CONCLUSIONS

We conclude that free-ranging dogs exhibit reduced object-neophobia in a scavenging context by significantly inspecting a relatively simple object with potential novelty more over familiar objects. Due to the use of only one type of novel object, we could not test its construct validity and subsequently failed to measure the extent of novelty. However, with adequate control and validation phases, we found that dogs indeed paid attention to the appearance of the novel object and made their foraging decisions. In future, it would be useful to include novel objects with different shapes and sizes to examine whether our results hold. Additionally, with our one-off tests (due to the between-subject study design), we could not check for repeatability and inter-individual differences in dogs' behavioural responses to novel objects. Such an approach would be instrumental in resolving whether object-neophobia is a personality (see *Sih, Bell & Johnson, 2004*) or a plastic (see *Vincze et al., 2016*; *Greggor et al., 2016*) trait. Nevertheless, we provide the first experimental evidence of free-ranging dogs' behavioural responses to novelty during scavenging.

## ACKNOWLEDGEMENTS

We thank Dr. Susnata Karmakar for allowing us to use his vehicle during fieldwork. We thank the Indian Institute of Science Education and Research Kolkata for providing infrastructural support.

### Funding

This work was supported by a grant from the SERB, Department of Science and Technology, Government of India (Project no. EMR/2016/000595). Debottam Bhattacharjee was supported by a DST INSPIRE PhD Fellowship, Department of Science and Technology, Government of India. Jayjit Das was supported by a DST INSPIRE Scholarship for Higher Education, Department of Science and Technology, Government of India. The funders had no role in study design, data collection and analysis, decision to publish, or preparation of the manuscript.

### Grant Disclosures

The following grant information was disclosed by the authors:
The SERB, Department of Science and Technology, Government of India: EMR/2016/000595.
DST INSPIRE PhD Fellowship.
Department of Science and Technology, Government of India.
DST INSPIRE Scholarship for Higher Education, Department of Science and Technology, Government of India.

### Competing Interests

Anindita Bhadra is an academic editor of PeerJ.

### Author Contributions

- Debottam Bhattacharjee conceived and designed the experiments, performed the experiments, analyzed the data, prepared figures and/or tables, authored or reviewed drafts of the article, decoding videos, and approved the final draft.
- Shubhra Sau performed the experiments, analyzed the data, prepared figures and/or tables, decoding videos, and approved the final draft.
- Jayjit Das performed the experiments, analyzed the data, prepared figures and/or tables, decoding videos, and approved the final draft.
- Anindita Bhadra conceived and designed the experiments, authored or reviewed drafts of the article, and approved the final draft.

### Animal Ethics

The following information was supplied relating to ethical approvals (i.e., approving body and any reference numbers):
Ethical approval of the study was obtained from Indian Institute of Science Education and Research Kolkata (Approval no. 1385/ac/10/CPCSEA).

### Field Study Permissions

The following information was supplied relating to field study approvals (i.e., approving body and any reference numbers):
Our work did not require field permit, as the experiments were conducted on streets, following the guidelines laid down by the Prevention of Cruelty to Animals Act 1960, Government of India.

## Data Availability

The raw data is available in the Supplemental File.

## Supplemental Information

Supplemental information for this article can be found online at http://dx.doi.org/10.7717/peerj.17121#supplemental-information.

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
