# Peer review of "Does novelty influence the foraging decisions of a scavenger?"

_PeerJ, doi:10.7717/peerj.17121_

## Round 0.1 · original submission · Minor Revisions

Overview
This study tested the hypothesis that free-ranging dogs would not exhibit neophobia by presenting them simultaneously with food items in plastic pouches (presumed familiar) and a plastic ball (presumed novel). On average, the dogs showed no avoidance of the ball, whether paired with a translucent or opaque plastic bag. Indeed, a majority of dogs inspected the ball first. When the translucent and opaque bags were paired, there was no difference in which was inspected first, indicating that visual cues of food did not influence first inspection. When balls with food and balls with only food odor were paired, there was also no difference, indicating that olfactory cues probably did not influence first inspection. The study therefore supported the hypothesis that opportunistic foraging and lack of major predation threats for free-ranging dogs would favor the lack of neophobia in free-ranging dogs.

The study is clear and well-written in general with the addition of helpful videos illustrating treatments and the dogs’ responses. Both reviewers consider it a useful addition to the literature but both also feel some additional clarifications are required, and Reviewer 2 suggests an additional analysis. Reviewer 2 provided numerous comments on the pdf, only partially overlapping with the their review. Therefore, in your rebuttal, please include all substantial comments not covered in the general comments (but not minor grammatical suggestions) from the pdf. I have also provided a pdf with suggestions on wording. I highlighted sections and then used comments to make suggestions for alternative wording. You do not have to include these comments in your rebuttal unless you disagree and have not made suggested changes. My comments below can be considered as a third review: make changes where the comments are valid and explain why if you do not find them valid.

Editor’s Comments
General Issues
Although it was only mentioned briefly, I think I understand how, in general, neophobia and neophilia are not simply two ends of a continuum, based on the underlying psychology. Reviewer 2 comments favorably on your decision to separate these concepts within the manuscript. However, I was not very clear on how this distinction relates to your work. For example, the abstract states that pet dogs exhibit neophilia before going on to state that your study examines neophobia. It is not clear to me how first approach to the ball is evidence against neophobia more than it is evidence in favor of neophilia. If there had been no difference in approaches to the ball and plastic bag, that would have been evidence against neophobia but not for neophilia; the higher rate of approaches to the ball suggests neophilia to me. I am not as familiar with this literature as the reviewers, so I might be wrong. Consider, however, that other readers might also not grasp this distinction and the implications of your findings; it is important that your conclusions be clear to a broad audience.
• Please clarify the issue of neophobia vs. neophilia in your Introduction and address it specifically with respect to your findings in the Discussion. It might help to specify how various possible outcomes would relate to the hypothesis.

In the Discussion, you might consider the implications of finding that 30-40% of the dogs went first to the plastic bag. Does this suggest that, while the majority of dogs do not show neophobia, a substantial number may? Is it possible that the generalization ‘free-ranging dogs do not show neophobia’ may be overstated?

You use ‘explorative’ a couple of times in the manuscript and ‘exploratory’ more frequently. Do you intend a distinct meaning for these terms? If not, use one term consistently. If you do intend a distinction, perhaps you should indicate what that distinction is.

The Instructions to Authors of PeerJ request that only the left margin be justified.

Comments on specific sections

Introduction
As noted above, I found the frequent switching between lack of neophobia and presence of neophilia in the literature review confusing throughout the Introduction. The information provided on the difference between these concepts was not sufficient for me to keep track of the issue or to understand how your study investigates lack of neophobia rather than presence of neophilia.

L183-185. The Introduction as not made it clear why you predict no difference in treatments c and d or how latency is related to the hypothesis. These issues should be clarified in the appropriate paragraphs.

Methods
L201. If you selected a dog truly randomly, you should have had a randomization procedure. If there was no specific procedure for selection, ‘haphazardly’ is a more appropriate term than ‘randomly’.
L214. Use metric units.
L214-216. Some clarification is needed here. Is a plastic pouch the same thing as North Americans would call a ‘plastic bag’ or something different? Was it sealed at the top (for example by a wire twist tie)? Can you determine how thick the plastic was because this could have affected transmission of odor? You write that the bag was translucent, which would imply that light gets through but vision of the contents was limited, rather than transparent, which would imply that it was made of clear plastic with contents very visible. If it was translucent rather than transparent, which might be the case as indicated by one of the videos, a brief mention of color, such as ‘a translucent white pouch’ and degree to which the food would be visible might help readers and anyone wanting to replicate your experiment. Indeed, you might include the brand and type of bag in the Methods.
L219ff. I agree with Reviewer 1 that it is challenging for readers to remember the treatments based on the similar acronyms. Even though more space would be required, replacing the acronyms with terms such as ball vs. opaque pouch, ball vs. translucent pouch, opaque vs. translucent pouch, and food vs. scent would greatly aid readability. You may be able to find alternative wording that would be as clear without being quite as long, but clarity to readers is the key. Similarly, when you refer to the ball or pouch in the text, you can write it out rather than abbreviate.
L264. Was latency measured only to the first object inspected? If so, please specify this. ‘An initial 2 m distance’ is not exactly clear. Do you mean that you estimated when the dog was 2 meters from the objects on the video and then measured the time between when the dog passed the 2 m distance and when it physically contacted the first object?
L266-268. It is unclear what you mean here. Please revise. Possibly, split into two or more sentences.

Discussion
The Discussion needs restructuring. The first paragraph summarizes the interpretations of your empirical results without the required step of first addressing the strength of the evidence and should be removed. The empirical finding that more dogs first approached the ball than first approached the pouch does not in itself show that the dogs are not neophobic. The lack of neophobia is an interpretation which requires a careful, critical analysis. The reviewers and I have suggested some of the issues that might be considered, for example absence of neophobia vs. presence of neophilia, whether the ball is truly novel, whether results might be related to only to balls because you used only a single unfamiliar stimulus, whether movement of the ball might influence the approach, and how to interpret the substantial number of dogs that went first to the pouch. Being much more familiar with the study, you can probably think of additional issues that should be addressed. The Discussion could start with a brief summary of the empirical findings regarding the stimulus approached first, then proceed to a careful argument about what conclusions can be drawn regarding neophobia followed by placing these findings in the context of previous literature. A subsequent paragraph could consider the findings of the two control treatments. Then you could address similarly the latency results.

Conclusions
The Conclusions should not be another repetition of the findings, but rather suggest broader implications of the study and fruitful potential questions for further research.

References
Unlike the rest of your manuscript, this section has numerous errors. Some journal article titles have capital letters throughout. One reference is duplicated. Some references lack volume and page numbers. In others, the information is out of order. I highlighted a few examples, but it is your responsibility to check all references carefully.

Table 1. Note that PeerJ instructions asks that you not embed tables in the text. As noted previously, the table would be clearer with words than with acronyms for the treatments. However, tables should not be redundant to the text. All or almost all of the information in Table i1 s also in the text. I think that careful rewriting of Methods would allow you to remove the table.

Fig. 1. I have made numerous suggestions regarding clarification of the figure caption. In addition, I suggest that you include the sample size (N = x) in parentheses under each treatment combination.

Reviewer 1 ·

Basic reporting

The manuscript is well written with sufficient background information and provided and literature cited. The raw data and R script are also provided. The figure and table look good, but I think the experimental conditions could be clarified a bit. I think it would be helpful to label the experimental conditions with a descriptive term in the table, results, and figure in order to help the reader more quickly understand what you are presenting. I understand that you describe the conditions in Table 1, but it’s hard to remember the acronyms when looking at results elsewhere and I think it would be helpful to have just a one or two word descriptor for each along with the acronym. So for example in Table 1, under experimental condition you could have GB-BP (novel vs opaque familiar), GB-TP (novel vs transparent familiar), etc. And write the same when presenting each result section so the reader doesn’t have to scroll up to the table to reference. I understand it may take up space in the figure to add these labels, so they could be included in the caption instead, whatever the authors think looks best. Overall I think this would help the reader keep track of what conditions are testing novelty, as while I was reading I often had to go back to the table to remind myself since the acronyms are similar.

Experimental design

The research question is relevant and understanding how free-ranging dogs reactions to novelty may influence their survival in urban environments is important. The field methodology seems appropriate, but some clarifications would help the reader better understand the experimental design.

First you could add some clarity about the study subjects. I understand you sampled across a wide area, but isn’t it possible that dogs could have moved between areas over the course of your study? Were there other ways that you were certain that the same dogs weren’t tested multiple times such as visual comparisons of your video data, etc? I think this needs to be described in a bit more detail or the possibility of subjects being tested in multiple conditions should be discussed.

It was a little unclear to me what the latency comparison was looking at in the methods so some clarification there would help. If I’m understanding correctly, I think that the latencies for dogs who first inspected one object were compared to the latencies for dogs who first inspected the other object. Your hypothesis was that they would have similar latencies to approach objects under different conditions (lines 184-185), but the comparison seems to be within a condition so I found this a bit confusing. Was your hypothesis that dogs who approach the familiar and those who approach novel objects first would do so with similar latencies if they were not neophobic?

Validity of the findings

The statistics are robust and the results are clear, however I have some comments about additional discussion that I think should be included to frame the study's conclusions.
I think you should include discussion about the limited novelty of the balls used as novel objects in this study and why a more extreme novel object wasn't chosen. The authors stated that balls are not prevalent in the environment, but I could imagine that some dogs would have experienced children’s balls or ball-like objects in an urban area. I think the dogs could have shown a greater neophobic response with a more complex or more novel object (see citations below) so some more discussion about the object choice should be included.
Mettke-Hofmann, C., Rowe, K. C., Hayden, T. J., & Canoine, V. (2006). Effects of experience and object complexity on exploration in garden warblers (Sylvia borin). Journal of Zoology, 268(4), 405–413. https://doi.org/10.1111/j.1469-7998.2005.00037.x
Greggor, A. L., Thornton, A., & Clayton, N. S. (2015). Neophobia is not only avoidance: Improving neophobia tests by combining cognition and ecology. Current Opinion in Behavioral Sciences, 6, 82–89. https://doi.org/10.1016/j.cobeha.2015.10.007

Do you think that the mobility of the ball could have influenced your results? I noticed in one video that after being placed it rolled and I wonder if this could have happened frequently and if that motion could be more attractive to the dogs. I don’t think this invalidates your results because it’s still a novel object, but I think it’s worth discussing this possibility.

In some places throughout the manuscript the dog’s initial inspection of an object is discussed as a preference, and while I understand that this is typical in an object-choice task, I started thinking about whether it’s demonstrating a “preference” in this case or are they inspecting the unknown object first to reduce uncertainty about any potential danger? Perhaps the inclusion of food in the objects supports the choice as a preference, but I think this aspect of the dogs' choices should be discussed in this context.

Additional comments

Some grammatical and clarification suggestions
49- delete “are”
157 clarify “early age classes”, juveniles or another descriptor would be more clear
160 suggest “previous studies have not investigated…”
166-177 I think it would be helpful for the reader to know before reading all the combinations of objects that the dogs received, that plastic balls were novel and pouches familiar objects since as I read through combinations I was questioning. I also think you could provide a more general summary in this paragraph rather than go into each condition, since those are presented in detail in the methods section.
171 suggest “by gently rubbing it with a piece of food reward.”
224 When in the experimental timeline did this change? Is it possible this influenced results?
255 suggest “the subject dogs only witnessed the experimenters briefly during this test.”
288-309 I think it would be more clear when presenting these results to write “adults” and "juveniles" after the corresponding numbers in parentheses. For example, "62% (23 out of 37 adults) and 69% (18 out of 26 juveniles) inspected GB first"
360 suggest adding “for dogs” before citation or “by dogs” after tolerated
361 I’m not sure what is meant by “contrary to test conditions” so suggest deleting
370-371 This sentence is a bit confusing because you say neophobia may still be relevant in contexts without food which you define as neophilia, so I'm left wondering which it is. I commend the authors on distinguishing between neophobia and neophilia throughout this manuscript, but this was one sentence where the two traits seemed jumbled. Perhaps replacing neophobia with more general “responses to novelty” will address this issue.
394 I think it would be helpful to clarify here that you’re comparing the latencies of the first choices by different dogs, so those that approached the novel object first didn’t do so faster than those that approached the familiar object first. The rest of the paragraph explains this a bit, but when reading I was questioning what latencies were compared here.

Reviewer 2 ·

Basic reporting

The overall structure of the research article and its elements conforms to the standard norms for scientific reporting and I found them to be elegantly drafted. Every section of the article is presented with adequate comprehensiveness though few paragraphs in the Introduction and the Discussion sections can be made succinct and redundancies in ideas and phrases can be removed. For example, in the Introduction section, 'neophobia' and 'neophilia' could be independently defined, and the methodologies to study the two should be clearly contrasted. Correspondingly, in the Discussion section, the interpretation of the results should be limited to neophobia and any comment made on neophilia should be labelled as speculative and readers should be appropriately cautioned. I reckon that the Discussion section will vastly benefit from comparison of free-ranging dogs with appropriate (such as foraging and neophobia) studies on urban canids, and I urge the authors to strongly consider it.

Experimental design

The research being reported is original and fills an identified knowledge gap. Regardless, there are serious shortcomings in rigor of the methodology.

Firstly, the experimental protocol for neophobia and neophilia is poorly described and not well established by the authors in their manuscript. Secondly, no justification is provided on why neophobia is given precedence for conducting this research over neophilia. Thirdly, no justification is provided for the choice of the novel object and its physical attributes. Fourthly, convergent validity and discriminant validity of the experimental protocol (i.e., to measure neophobia) is not established. Fifthly, recommended guidelines suggest the use of at least two equivalent types of novel objects in neophobia research for avoiding pseudoreplication, whereas just one kind was used in the study. Sixthly, the authors have not validated the use of green plastic balls as being novel, which is a key step in such research. Seventhly, I would suggest the authors to statistically check for any effect of time of conducting the experiment, a proxy for satiety (hunger) on choice and latency of approach.

Validity of the findings

The authors have been thorough in describing the data, and analyzing them. Apart from the suggested statistical test, the analyses performed by the authors are well justified and supports their proposed hypothesis. The findings are statistically supported and appropriately controlled. The results align with the hypotheses tested. However, the finding that the latencies of approach were comparable across test conditions, i.e., between novel and familiar food-baited objects is not amply justified. I would urge the authors to provide stronger arguments. Additionally, the authors appear to over-generalize the implication of their study of 'object neophobia' on foraging decisions in the Discussion section. The authors should distinguish object neophobia from other unstudied contexts in which neophobia operates such as novel food and novel environment, and exercise caution while describing the scope of their finding on general foraging decisions by free-ranging dogs. Correspondingly, appropriate sections of Discussion need revision (e.g. Line 379-382).

Additional comments

Additional reviews, edits and comments are provided along with the main manuscript.

Annotated reviews are not available for download in order to protect the identity of reviewers who chose to remain anonymous.

---

## Round 0.2 · accepted · Accept

The authors have made appropriate changes and, in a few cases, explained why they did not follow reviewers' suggestions. I consider the manuscript now ready for the publication process, except for a small number of corrections that can be made during the processing of the manuscript. I have highlighted those changes in the attached pdf.